# Comparator-Adaptive Convex Bandits

**Dirk van der Hoeven**
Mathematical Institute
Leiden University
dirk@dirkvanderhoeven.com

**Ashok Cutkosky**
Boston University
ashok@cutkosky.com

**Haipeng Luo**
Computer Science Department
University of Southern California
haipengl@usc.edu

## Abstract

We study bandit convex optimization methods that adapt to the norm of the comparator, a topic that has only been studied before for its full-information counterpart. Specifically, we develop convex bandit algorithms with regret bounds that are small whenever the norm of the comparator is small. We first use techniques from the full-information setting to develop comparator-adaptive algorithms for linear bandits. Then, we extend the ideas to convex bandits with Lipschitz or smooth loss functions, using a new variant of the standard single-point gradient estimator and carefully designed surrogate losses.

## 1 Introduction

In many situations, information is readily available. For example, if a gambler were to bet on the outcome of a football game, he can observe the outcome of the game regardless of what bet he made. In other situations, information is scarce. For example, the gambler could be deciding what to eat for dinner: should I eat a salad, a pizza, a sandwich, or not at all? These actions will result in different and unknown outcomes, but the gambler will only see the outcome of the action he actually takes, with one notable exception: not eating results in a predetermined outcome of being very hungry.

These two situations are instantiations of two different settings in online convex optimization: the full information setting and the bandit setting. More formally, both settings are sequential decision making problems where in each round $t = 1, \ldots, T$, a learner has to make a prediction $\boldsymbol{w}_t \in \mathcal{W} \subseteq \mathbb{R}^d$ and an adversary provides a convex loss function $\ell_t : \mathcal{W} \to \mathbb{R}$. Afterwards, in the full information setting [27] the learner has access to the loss function $\ell_t$, while in the bandit setting [19, 13] the learner only receives the loss evaluated at the prediction, that is, $\ell_t(\boldsymbol{w}_t)$. In both settings the goal is to minimize the regret with respect to some benchmark point $\boldsymbol{u}$ in hindsight, referred to as the *comparator*. More specifically, the regret against $\boldsymbol{u}$ is the difference between the total loss incurred by the predictions of the learner and that of the comparator:

$$\mathcal{R}_T(\boldsymbol{u}) = \sum_{t=1}^{T} \ell_t(\boldsymbol{w}_t) - \ell_t(\boldsymbol{u}).$$

When the learner's strategy is randomized, we measure the performance by the expected regret $\mathbb{E}[\mathcal{R}_T(\boldsymbol{u})]$.

Standard algorithms in both the full information setting and the bandit setting assume that the learner's decision space $\mathcal{W}$ is a convex *compact* set and achieve sublinear regret against the optimal comparator in this set: $\boldsymbol{u} = \arg\min_{\boldsymbol{u}^* \in \mathcal{W}} \sum_{t=1}^{T} \ell_t(\boldsymbol{u}^*)$. To tune these standard algorithms optimally, however, one requires knowledge of the norm of the comparator $\|\boldsymbol{u}\|$, which is unknown. A common work-around is to simply tune the algorithms in terms of the worst-case norm: $\max_{\boldsymbol{u} \in \mathcal{W}} \|\boldsymbol{u}\|$, assumed to be 1 without loss of generality. This results in worst-case bounds that do not take advantage of the case when $\|\boldsymbol{u}\|$ is small. For example, when the loss functions are $L$-Lipschitz, classic Online

Table 1: Summary of main results. Regret is measured with respect to the total loss of an arbitrary point $\boldsymbol{u} \in \mathbb{R}^d$ in the unconstrained setting, or an arbitrary point $\boldsymbol{u} \in \mathcal{W}$ in the constrained setting with a decision space $\mathcal{W}$ contained in the unit ball. $T$ is the total number of rounds, $1/c$ is radius of the largest ball contained by $\mathcal{W}$, and $\nu$ is the self-concordant parameter. Both $c$ and $\nu$ are bounded by $O(d)$.

| Loss functions ($L$-Lipschitz) | Regret for unconstrained settings | Regret for constrained settings |
|---|---|---|
| Linear (Section 3.2) | $\widetilde{O}\left(\|\boldsymbol{u}\| dL\sqrt{T}\right)$ | $\widetilde{O}\left(\|\boldsymbol{u}\| cdL\sqrt{T}\right)$ |
| Convex (Section 4.1 and 4.2) | $\widetilde{O}\left(\|\boldsymbol{u}\| L\sqrt{d}T^{\frac{3}{4}}\right)$ | $\widetilde{O}\left(\|\boldsymbol{u}\| cL\sqrt{d}T^{\frac{3}{4}}\right)$ |
| Convex and $\beta$-smooth (Section 4.2) | $\widetilde{O}\left(\max\{\|\boldsymbol{u}\|^2, \|\boldsymbol{u}\|\}\beta(dLT)^{\frac{2}{3}}\right)$ | - |

Gradient Descent [27] guarantees $\mathcal{R}_T(\boldsymbol{u}) = O(L\sqrt{T})$ in the full information setting, while the algorithm of [13] guarantees $\mathbb{E}\left[\mathcal{R}_T(\boldsymbol{u})\right] = O(d\sqrt{L}T^{3/4})$ in the bandit setting, both of which are independent of $\|\boldsymbol{u}\|$.

Recently, there has been a series of works in the full information setting that addresses this problem by developing *comparator-adaptive* algorithms, whose regret against $\boldsymbol{u}$ depends on $\|\boldsymbol{u}\|$ for *all* $\boldsymbol{u} \in \mathcal{W}$ *simultaneously* (see for example McMahan and Orabona [22], Orabona and Pál [23], Foster et al. [14], Cutkosky and Boahen [9], Kotlowski [20], Cutkosky and Orabona [10], Foster et al. [16], Jun and Orabona [18], Van der Hoeven [25]). These bounds are often not worse than the standard worst-case bounds, but could be much smaller in the case when there exists a comparator with small norm and reasonably small total loss. Moreover, most of these results also hold for the so-called *unconstrained* setting where $\mathcal{W} = \mathbb{R}^d$, that is, both the learner's predictions and the comparator can be any point in $\mathbb{R}^d$. For example, Cutkosky and Orabona [10] achieve $\mathcal{R}_T(\boldsymbol{u}) = \widetilde{O}(\|\boldsymbol{u}\| L\sqrt{T})$ for all $\boldsymbol{u}$, in both the constrained and unconstrained settings, under full information feedback.[1]

While developing comparator-adaptive algorithms is relatively well-understood at this point in the full information setting, to the best of our knowledge, this has not been studied at all for the more challenging bandit setting. In this work, we make the first attempt in this direction and develop comparator-adaptive algorithms for several situations, including learning with linear losses, general convex losses, and convex and smooth losses, for both the constrained and unconstrained settings. Our results are summarized in Table 1. Ignoring other parameters for simplicity, for the linear case, we achieve $\widetilde{O}(\|\boldsymbol{u}\|\sqrt{T})$ regret (Section 3.2); for the general convex case, we achieve $\widetilde{O}(\|\boldsymbol{u}\| T^{\frac{3}{4}})$ regret in both the constrained and unconstrained setting (Sections 4.1 and 4.2); and for the convex and smooth case, we achieve $\widetilde{O}\left(\max\{\|\boldsymbol{u}\|^2, \|\boldsymbol{u}\|\}\beta(dLT)^{\frac{2}{3}}\right)$ regret in the unconstrained setting (Section 4.1).

In order to achieve our results for the convex case, we require an assumption on the loss, namely that the value of $\ell_t(\mathbf{0})$ is known for all $t$.[2] While restrictive at first sight, we believe that there are abundant applications where this assumption holds. As one instance, in control or reinforcement learning problems, $\mathbf{0}$ may represent some nominal action which has a known outcome: not eating results in hunger, or buying zero inventory will result in zero revenue. Another application is a classification problem where the features are not revealed to the learner. For example, end-users of a prediction service may not feel comfortable revealing their information to the service. Instead, they may be willing to do some local computation and report the loss of the service's model. Most classification models (e.g. logistic regression) have the property that the loss of the $\mathbf{0}$ parameter is a known constant regardless of the data, and so this situation would also fit into our framework. Common loss functions that satisfy this assumption are linear loss, logistic loss, and hinge loss.

**Techniques** Our algorithms are based on sophisticated extensions of the black-box reduction introduced by Cutkosky and Orabona [10], which separately learns the magnitude and the direction of the prediction. To make the reduction work in the bandit setting, however, new ideas are required, including designing an appropriate surrogate loss function and a new variant of the standard one-point gradient estimator with time-varying parameters. Note that Cutkosky and Orabona [10] also propose

a method to convert any unconstrained algorithm to a constrained one in the full information setting, but this does not work in the bandit setting for technical reasons. Instead, we take a different approach by constraining the magnitude of the prediction directly.

**Related work**    As mentioned, there has been a line of recent work on comparator-adaptive algorithms for the full information setting. Most of them do not transfer to the bandit setting, except for the approach of Cutkosky and Orabona [10] from which we draw heavy inspiration. To the best of our knowledge, comparator-adaptive bandit algorithms have not been studied before. Achieving "adaptivity" in a broader sense is generally hard for problems with bandit feedback; see negative results such as [12, 21] as well as recent progress such as [7, 15].

In terms of worst-case (non-adaptive) regret, the seminal work of [1] is the first to achieve $O(\sqrt{T})$ regret for bandit with linear losses, and [19, 13] are the first to achieve sublinear regret for general convex case. Over the past decade, the latter result has been improved in many different ways [2, 24, 3, 17], and regret of order $O(\sqrt{T})$ under no extra assumptions was recently achieved [4, 5, 6]. However, these $O(\sqrt{T})$ bounds are achieved by very complicated algorithms that incur a huge dependence on the dimension $d$. Our algorithms are more aligned with the simpler ones with milder dimension-dependence [1, 13, 24] and achieve the same dependence on $T$ in different cases. How to achieve comparator-adaptive regret of order $O(\sqrt{T})$ for the general convex case is an important future direction.

## 2    Preliminaries

In this section, we describe our notation, state the definitions we use, and introduce the bandit convex optimization setting formally. We also describe the black-box reduction of [10] we will use throughout the paper.

**Notation and definitions**    The inner product between vectors $\boldsymbol{g} \in \mathbb{R}^d$ and $\boldsymbol{w} \in \mathbb{R}^d$ is denoted by $\langle \boldsymbol{w}, \boldsymbol{g} \rangle$. $\mathbb{R}_+$ denotes the set of positive numbers. The Fenchel conjugate $F^\star$ of a convex function $F$ is defined as $F^\star(\boldsymbol{w}) = \sup_{\boldsymbol{g}} \langle \boldsymbol{w}, \boldsymbol{g} \rangle - F(\boldsymbol{g})$. $\| \cdot \|$ denotes a norm and $\|\boldsymbol{g}\|_\star = \sup_{\boldsymbol{w}:\|\boldsymbol{w}\| \leq 1} \langle \boldsymbol{w}, \boldsymbol{g} \rangle$ denotes the dual norm of $\boldsymbol{g}$. The Bregman divergence associated with convex function $F$ between points $\boldsymbol{x}$ and $\boldsymbol{y}$ is denoted by $B_F(\boldsymbol{x}\|\boldsymbol{y}) = F(\boldsymbol{x}) - F(\boldsymbol{y}) - \langle \nabla F(\boldsymbol{y}), \boldsymbol{x} - \boldsymbol{y} \rangle$, where $\nabla F(\boldsymbol{x})$ denotes the gradient of $F$ evaluated at $\boldsymbol{x}$. The unit ball equipped with norm $\|\cdot\|$ is denoted by $\mathcal{B} = \{\boldsymbol{w} : \|\boldsymbol{w}\| \leq 1\}$. The unit sphere with norm $\| \cdot \|$ is denoted by $\mathcal{S} = \{\boldsymbol{w} : \|\boldsymbol{w}\| = 1\}$. The unit ball and sphere with norm $\| \cdot \|_2$ are denoted by $\mathbb{B}$ and $\mathbb{S}$ respectively. $\boldsymbol{x} \sim U(\mathcal{Z})$ denotes that $\boldsymbol{x}$ follows the uniform distribution over $\mathcal{Z}$. We say a function $f$ is $\beta$-smooth over the set $\mathcal{W}$ if the following holds:

$$f(\boldsymbol{y}) \leq f(\boldsymbol{x}) + \langle \nabla f(\boldsymbol{x}), \boldsymbol{y} - \boldsymbol{x} \rangle + \frac{\beta}{2} \|\boldsymbol{x} - \boldsymbol{y}\|_2^2, \quad \forall \boldsymbol{x}, \boldsymbol{y} \in \mathcal{W}.$$

We say a function $f$ is $L$-Lipschitz over the set $\mathcal{W}$ if the following holds:

$$|f(\boldsymbol{y}) - f(\boldsymbol{x})| \leq L\|\boldsymbol{y} - \boldsymbol{x}\|_2, \quad \forall \boldsymbol{x}, \boldsymbol{y} \in \mathcal{W}.$$

Throughout the paper we will assume that $\beta, L \geq 1$. Also, by mild abuse of notation, we use $\partial f(x)$ to indicate an arbitrary subgradient of a convex function $f$ at $x$.

All of our algorithms are reductions that use prior algorithms in disparate ways to obtain our new results. In order for these reductions to work, we need some assumptions on the base algorithms. We will encapsulate these assumptions in *interfaces* that describe inputs, outputs, and guarantees described by an algorithm rather than its actual operation (see Interfaces 3 and 4 for examples). We can use specific algorithms from the literature to implement these interfaces, but our results depend only on the properties described in the interfaces.

### 2.1    Bandit Convex Optimization

The bandit convex optimization protocol proceeds in rounds $t = 1, \dots, T$. In each round $t$ the learner plays $\boldsymbol{w}_t \in \mathcal{W} \subseteq \mathbb{R}^d$. Simultaneously, the environment picks an $L$-Lipschitz convex loss function $\ell_t : \mathcal{W} \to \mathbb{R}$, after which the learner observes $\ell_t(\boldsymbol{w}_t)$. Importantly, the learner only observes the loss function evaluated at $\boldsymbol{w}_t$, not the function itself. This forces the learner to play random points and

---
**Algorithm 1** Black-Box Reduction with Full Information
---
1: **Input:** "Direction" algorithm $\mathcal{A}_{\mathcal{Z}}$ and "scaling" algorithm $\mathcal{A}_{\mathcal{V}}$
2: **for** $t = 1 \dots T$ **do**
3:    Get $\boldsymbol{z}_t \in \mathcal{Z}$ from $\mathcal{A}_{\mathcal{Z}}$
4:    Get $v_t \in \mathbb{R}$ from algorithm $\mathcal{A}_{\mathcal{V}}$
5:    Play $\boldsymbol{w}_t = v_t \boldsymbol{z}_t$, receive $\boldsymbol{g}_t$
6:    Send $\boldsymbol{g}_t$ to algorithm $\mathcal{A}_{\mathcal{Z}}$ as the $t$-th loss vector
7:    Send $\langle \boldsymbol{z}_t, \boldsymbol{g}_t \rangle$ to algorithm $\mathcal{A}_{\mathcal{V}}$ as the $t$-th loss value
8: **end for**
---

estimate the feedback he wants to use to update $\boldsymbol{w}_t$. Therefore, in the bandit feedback setting, the goal is to bound the *expected* regret $\mathbb{E}\left[\mathcal{R}_T(\boldsymbol{u})\right]$, where the expectation is with respect to both the learner and the environment.

We make a distinction between linear bandits, where $\ell_t(\boldsymbol{w}) = \langle \boldsymbol{w}, \boldsymbol{g}_t \rangle$, and convex bandits, where $\ell_t$ can be any $L$-Lipschitz convex function. Throughout the paper, if $\mathcal{W} \neq \mathbb{R}^d$ we assume that $\mathcal{W}$ is compact, has a non-empty interior, and contains $\boldsymbol{0}$. Without loss of generality we assume that $\frac{1}{c}\mathbb{B} \subseteq \mathcal{W} \subseteq \mathbb{B}$ for some $c \geq 1$. Some of our bounds depend on $c$, which, without loss of generality, is always bounded by $d$, due to a reshaping trick discussed in [13].

## 2.2 Black-Box Reductions with Full Information

Our algorithms are based on a black-box reduction from [10] for the full information setting (see Algorithm 1). The reduction works as follows. In each round $t$ the algorithms plays $\boldsymbol{w}_t = v_t \boldsymbol{z}_t$, where $\boldsymbol{z}_t \in \mathcal{Z}$ for some domain $\mathcal{Z}$, is the prediction of a constrained algorithm $\mathcal{A}_{\mathcal{Z}}$, and $v_t$ is the prediction of a one-dimensional algorithm $\mathcal{A}_{\mathcal{V}}$. The goal of $\mathcal{A}_{\mathcal{Z}}$ is to learn the direction of the comparator while the goal of $\mathcal{A}_{\mathcal{V}}$ is to learn the norm of the comparator. Let $\boldsymbol{g}_t$ be the gradient of $\ell_t$ at $\boldsymbol{w}_t$, which is known to the algorithm in the full information setting. We feed $\boldsymbol{g}_t$ as feedback to $\mathcal{A}_{\mathcal{Z}}$ and $\langle \boldsymbol{z}_t, \boldsymbol{g}_t \rangle$ as feedback to $\mathcal{A}_{\mathcal{V}}$. Although the original presentation considers only $\mathcal{Z} = \mathcal{B}$, we will need to extend the analysis to more general domains.

As outlined by Cutkosky and Orabona [10], the regret of Algorithm 1 decomposes into two parts. The first part of the regret is for learning the norm of $\boldsymbol{u}$, and is controlled by Algorithm $\mathcal{A}_{\mathcal{V}}$. The second part of the regret is for learning the direction of $\boldsymbol{u}$ and is controlled by $\mathcal{A}_{\mathcal{Z}}$. The proof is provided in Appendix A for completeness.

**Lemma 1.** *Let* $\mathcal{R}_T^{\mathcal{V}}(\|\boldsymbol{u}\|) = \sum_{t=1}^{T}(v_t - \|\boldsymbol{u}\|)\langle \boldsymbol{z}_t, \boldsymbol{g}_t \rangle$ *be the regret for learning* $\|\boldsymbol{u}\|$ *by Algorithm* $\mathcal{A}_{\mathcal{V}}$ *and let* $\mathcal{R}_T^{\mathcal{Z}}\left(\frac{\boldsymbol{u}}{\|\boldsymbol{u}\|}\right) = \sum_{t=1}^{T}\langle \boldsymbol{z}_t - \frac{\boldsymbol{u}}{\|\boldsymbol{u}\|}, \boldsymbol{g}_t \rangle$ *be the regret for learning* $\frac{\boldsymbol{u}}{\|\boldsymbol{u}\|}$ *by* $\mathcal{A}_{\mathcal{Z}}$. *Then Algorithm 1 satisfies*

$$\mathcal{R}_T(\boldsymbol{u}) = \mathcal{R}_T^{\mathcal{V}}(\|\boldsymbol{u}\|) + \|\boldsymbol{u}\|\mathcal{R}_T^{\mathcal{Z}}\left(\frac{\boldsymbol{u}}{\|\boldsymbol{u}\|}\right). \tag{1}$$

Cutkosky and Orabona [10] provide an algorithm to ensure $\mathcal{R}_T^{\mathcal{V}}(\|\boldsymbol{u}\|) = \widetilde{O}\left(1 + \|\boldsymbol{u}\|L\sqrt{T}\right)$, given that $\|\boldsymbol{g}_t\|_\star \leq L$. This algorithm satisfies the requirements described later in Interface 3, and will be used throughout this paper.

# 3 Comparator-Adaptive Linear Bandits

Now, we apply the reduction of section 2.2 to develop comparator-adaptive algorithms for linear bandits. We will see that in the unconstrained case, the reduction works almost without modification, but in the constrained case we will need to be more careful to enforce the constraints.

## 3.1 Unconstrained Linear Bandits

We begin by discussing the unconstrained linear bandit setting, which turns out to be the easiest setting we consider. Following Algorithm 1, we will still play $\boldsymbol{w}_t = v_t \boldsymbol{z}_t$. However, instead of taking a fixed $\boldsymbol{z}_t$ from a full-information algorithm, we take a random $\boldsymbol{z}_t$ from a *bandit* algorithm.

---

**Algorithm 2** Black-Box Reduction for Linear Bandits

---

1: **Input:** Constrained Linear Bandit Algorithm $\mathcal{A}_{\mathcal{Z}}$ and unconstrained 1-d Algorithm $\mathcal{A}_{\mathcal{V}}$
2: **for** $t = 1 \dots T$ **do**
3:    Get $\boldsymbol{z}_t \in \mathcal{Z}$ from $\mathcal{A}_{\mathcal{Z}}$
4:    Get $v_t \in \mathbb{R}$ from $\mathcal{A}_{\mathcal{V}}$
5:    Play $\boldsymbol{w}_t = v_t \boldsymbol{z}_t$
6:    Receive loss $\langle \boldsymbol{w}_t, \boldsymbol{g}_t \rangle$
7:    Compute $\mathcal{L}_t = \frac{1}{v_t} \langle \boldsymbol{w}_t, \boldsymbol{g}_t \rangle = \langle \boldsymbol{z}_t, \boldsymbol{g}_t \rangle$.
8:    Send $\mathcal{L}_t$ to Algorithm $\mathcal{A}_{\mathcal{Z}}$ as $t$-th loss value.
9:    Send $\mathcal{L}_t$ to Algorithm $\mathcal{A}_{\mathcal{V}}$ as $t$-th loss value.
10: **end for**

---

**Interface 3** Scale Learning Interface (see example implementation in [10])

---

1: **Input:** A line segment $l \subseteq \mathbb{R}$
2: **for** $t = 1 \dots T$ **do**
3:    Play $v_t \in l$
4:    Receive loss value $g_t$ such that $|g_t| \leq L_{\mathcal{V}}$
5: **end for**
6: **Ensure:** for all $\hat{v} \in l$, $\sum_{t=1}^{T} (v_t - \hat{v}) g_t = \widetilde{O}\left( 1 + |\hat{v}| L_{\mathcal{V}} \sqrt{T} \right)$

---

Importantly, we can recover $\langle \boldsymbol{z}_t, \boldsymbol{g}_t \rangle$ exactly since $\langle \boldsymbol{w}_t, \boldsymbol{g}_t \rangle \frac{1}{v_t} = \langle \boldsymbol{z}_t, \boldsymbol{g}_t \rangle$. This means that we have enough information to send appropriate feedback to both $\mathcal{A}_{\mathcal{V}}$ and $\mathcal{A}_{\mathcal{Z}}$ and apply the argument of Lemma 1. Interestingly, we use a full-information one-dimensional algorithm for $\mathcal{A}_{\mathcal{V}}$, and only need $\mathcal{A}_{\mathcal{Z}}$ to take bandit input. This is because $\mathcal{A}_{\mathcal{V}}$ gets full information in the form of $\langle \boldsymbol{z}_t, \boldsymbol{g}_t \rangle$.

The algorithm $\mathcal{A}_{\mathcal{Z}}$ for learning the direction, on the other hand, now must be a bandit algorithm because intuitively we do not immediately get the full direction information $\boldsymbol{g}_t$ from the value of the loss alone. We will need this algorithm to fulfil the requirements described by Interface 4. One such algorithm is given by continuous Exponential Weights on a constrained set (see Van der Hoeven et al. [26, section 6] for details).

Our unconstrained linear bandit algorithm then is constructed from Algorithm 2 by choosing an algorithm that implements Interface 4 as $\mathcal{A}_{\mathcal{Z}}$ and Interface 3 with $l = \mathbb{R}$ as $\mathcal{A}_{\mathcal{V}}$. Plugging in the guarantees of the individual algorithms and taking the expectation of (1), the total expected regret is $\widetilde{O}(1 + \|\boldsymbol{u}\| dL \sqrt{T})$. Compared to the full information setting we have gained a factor $d$ in the regret bound, which is unavoidable given the bandit feedback [11]. The formal result is below.

**Theorem 1.** *Suppose $\mathcal{A}_{\mathcal{Z}}$ implements Interface 4 with domain $\mathcal{Z} = \mathcal{B}$ and $\mathcal{A}_{\mathcal{V}}$ implements Interface 3 with $l = \mathbb{R}_+$. Then Algorithm 2 satisfies for all $\boldsymbol{u} \in \mathbb{R}^d$:*

$$\mathbb{E}[\mathcal{R}(\boldsymbol{u})] = \widetilde{O}(1 + \|\boldsymbol{u}\| dL \sqrt{T}).$$

### 3.2 Constrained Linear Bandits

The algorithm in the previous section only works for $\mathcal{W} = \mathbb{R}^d$. In this section, we consider a compact set $\mathcal{W} \subset \mathbb{R}^d$.

In the full-information setting, Cutkosky and Orabona [10] provide a projection technique for producing constrained algorithms from unconstrained ones. Unfortunately, this technique does not translate directly to the bandit setting, and we must be more careful in designing our constrained linear bandit algorithm. The key idea is to constrain the internal scaling algorithm $\mathcal{A}_{\mathcal{V}}$, rather than attempting to constrain the final predictions $\boldsymbol{w}_t$. Enforcing constraints on the scaling algorithm's outputs $v_t$ will naturally translate into a constraint on the final predictions $\boldsymbol{w}_t$.

To produce a constrained linear bandit algorithm, we again use Algorithm 2, but now we instantiate $\mathcal{A}_{\mathcal{V}}$ implementing Interface 3 with $l = [0, 1]$ rather than $l = \mathbb{R}_+$, and instantiate $\mathcal{A}_{\mathcal{Z}}$ implementing Interface 4 with $\mathcal{Z} = \mathcal{W}$ rather than $\mathcal{Z} = \mathcal{B}$. As in the unconstrained setting, this allows us to feed full information feedback to $\mathcal{A}_{\mathcal{V}}$. At the same time restricting Interface 3 to $l = [0, 1]$ also guarantees

**Interface 4** Direction Learning Interface for Linear Bandits (see example implementation in [26])

1: **Input:** Domain $\mathcal{Z}$
2: **for** $t = 1 \ldots T$ **do**
3:      Play $\boldsymbol{z}_t \in \mathcal{Z}$
4:      Receive loss value $\langle \boldsymbol{z}_t, \boldsymbol{g}_t \rangle$ such that $|\langle \boldsymbol{z}_t, \boldsymbol{g}_t \rangle| \leq L$
5: **end for**
6: **Ensure:** for all $\boldsymbol{u} \in \mathcal{Z}$, $\mathbb{E}\left[ \sum_{t=1}^{T} \langle \boldsymbol{z}_t - \boldsymbol{u}, \boldsymbol{g}_t \rangle \right] = \widetilde{O}\left( dL\sqrt{T} \right)$

---

that $\boldsymbol{w}_t \in \mathcal{W}$. The regret bound of this algorithm is given in Theorem 2. The proof follows from combining Lemma 1 with the guarantees of Interfaces 3 and 4 and can be found in Appendix B.

**Theorem 2.** *Suppose $\mathcal{A}_{\mathcal{Z}}$ implements 4 with domain $\mathcal{Z} = \mathcal{W}$ and $\mathcal{A}_{\mathcal{V}}$ implements Interface 3 with $l = [0, 1]$. Then Algorithm 2 satisfies for all $\boldsymbol{u} \in \mathcal{W}$,*

$$\mathbb{E}[\mathcal{R}_T(\boldsymbol{u})] = \widetilde{O}\left( 1 + \|\boldsymbol{u}\| cdL\sqrt{T} \right).$$

If $\mathcal{W}$ is a unit ball, then $c = 1$. For other shapes of $\mathcal{W}$, recall that $c$ is at most $d$, which leads to a regret bound of $O\left( 1 + \|\boldsymbol{u}\| d^2 L\sqrt{T} \right)$.

## 4 Comparator-Adaptive Convex Bandits

In the general convex bandit problem, it is not clear how to use the single evaluation point feedback $\ell_t(\boldsymbol{w}_t)$ to derive any useful information about $\ell_t$. Fortunately, Flaxman et al. [13] solved this problem by using randomness to extract the gradients of a smoothed version of $\ell_t$. To adapt to the norm of the comparator, we employ the following tweaked version of smoothing used by Flaxman et al. [13]:

$$\ell_t^v(\boldsymbol{w}) = \mathbb{E}_{\boldsymbol{b} \sim U(\mathbb{B})}[\ell_t(\boldsymbol{w} + v\delta\boldsymbol{b})], \tag{2}$$

where $v, \delta > 0$. In contrast to prior work using this framework, our smoothing now depends on the scaling parameter $v$. Lemma 2 gives the gradient of $\ell_t^v(\boldsymbol{w})$ and is a straightforward adaptation of Lemma 2.1 by Flaxman et al. [13].

**Lemma 2.** *For $\delta \in (0, 1]$, $v > 0$:*

$$\nabla \ell_t^v(\boldsymbol{w}) = \frac{d}{v\delta} \mathbb{E}_{\boldsymbol{s} \sim U(\mathbb{S})}[\ell_t(\boldsymbol{w} + v\delta\boldsymbol{s})\boldsymbol{s}]. \tag{3}$$

With this lemma, we can estimate the gradient of the smoothed version of $\ell_t$ by evaluating $\ell_t$ at a random point, essentially converting the convex problem to a linear problem, except that one also needs to control the bias introduced by smoothing. Note that this estimate scales with $\frac{1}{v}$, which can be problematic if $v$ is small. To deal with this issue, we require one extra assumption: the value of $\ell_t(\boldsymbol{0})$ is known to the learner. As discussed in section 1, this assumption holds for several applications, including some control or reinforcement learning problems, where $\boldsymbol{0}$ represents a nominal action with a known outcome. Furthermore, certain loss functions satisfy the second assumption by default, such as linear loss, logistic loss, and hinge loss. Without loss of generality we assume that $\ell_t(\boldsymbol{0}) = 0$, as we can always shift $\ell_t$ without changing the regret.

Our general algorithm template is provided in Algorithm 5. It incorporates the ideas of Algorithm 2, but adds new smoothing and regularization elements in order to deal with the present more general situation. More specifically, it again makes use of subroutine $\mathcal{A}_{\mathcal{V}}$, which learns the scaling. The direction is learned by Online Gradient Descent [27], as was also done by Flaxman et al. [13]. Given $\boldsymbol{z}_t$ and $v_t$, our algorithm plays the point $\boldsymbol{w}_t = v_t(\boldsymbol{z}_t + \delta\boldsymbol{s}_t)$ for some parameter $\delta$ and $\boldsymbol{s}_t$ drawn uniformly at random from $\mathbb{S}$. By equation (3), we have

$$\mathbb{E}\left[ \frac{d}{v_t\delta} \ell_t(\boldsymbol{w}_t)s_t \right] = \nabla \ell_t^{v_t}(v_t\boldsymbol{z}_t). \tag{4}$$

This means that we can use $\hat{\boldsymbol{g}}_t = \frac{d}{v_t\delta} \ell_t(\boldsymbol{w}_t)s_t$ as an approximate gradient estimate, and we send this $\hat{\boldsymbol{g}}_t$ to Online Gradient Descent as the feedback. In other words, Online Gradient Descent itself is

---
**Algorithm 5** Black-Box Comparator-Adaptive Convex Bandit Algorithm
---
1: **Input:** Scaling algorithm $\mathcal{A}_{\mathcal{V}}$, $\delta \in (0, 1]$, $\alpha \in [0, 1]$, domain $\mathcal{Z} \subseteq \mathbb{B}$, and learning rate $\eta$
2: Set $z_1 = \mathbf{0}$
3: **for** $t = 1 \ldots T$ **do**
4:     Get $v_t$ from $\mathcal{A}_{\mathcal{V}}$
5:     Sample $s_t \sim U(\mathbb{S})$
6:     Set $w_t = v_t(z_t + \delta s_t)$
7:     Play $w_t$
8:     Receive $\ell_t(w_t)$
9:     Set $\hat{g}_t = \frac{d}{v_t \delta} \ell_t(w_t) s_t$
10:     **if** $\ell_t$ is $\beta$-smooth **then**
11:         Set $\bar{\ell}_t(v) = v\langle z_t, \hat{g}_t \rangle + \beta \delta^2 v^2$
12:     **else**
13:         Set $\bar{\ell}_t(v) = v\langle z_t, \hat{g}_t \rangle + 2\delta L|v|$
14:     **end if**
15:     Send $\partial \bar{\ell}_t(v_t)$ to algorithm $\mathcal{A}_{\mathcal{V}}$ as the $t$-th loss value
16:     Update $z_{t+1} = \arg\min_{z \in (1-\alpha)\mathcal{Z}} \eta\langle z, \hat{g}_t \rangle + \|z_t - z\|_2^2$
17: **end for**
---

essentially dealing with a full-information problem with gradient feedback and is required to ensure a regret bound $\mathbb{E}[\sum_{t=1}^T \langle z_t - u, \hat{g}_t \rangle] = \widetilde{O}(\frac{dL}{\delta}\sqrt{T})$ for all $u$ in some domain $\mathcal{Z}$. For technical reasons, we will also need to enforce $z_t \in (1 - \alpha)\mathcal{Z}$ for some $\alpha \in [0, 1]$. This restriction will be necessary in the constrained setting to ensure $v_t(z_t + \delta s_t) \in \mathcal{W}$.

Next, to specify the feedback to the scaling learning black-box $\mathcal{A}_{\mathcal{V}}$, we define a surrogate loss function $\bar{\ell}_t(v)$ which contains a linear term $v\langle z_t, \hat{g}_t \rangle$ and also a regularization term (see Algorithm 5 for the exact definition). The feedback to $\mathcal{A}_{\mathcal{V}}$ is then $\partial \bar{\ell}_t(v_t)$. Therefore, $\mathcal{A}_{\mathcal{V}}$ is essentially learning these surrogate losses, also with full gradient information. The regularization term is added to deal with the bias introduced by smoothing. This term does not appear in prior work on convex bandits, and it is one of the key components needed to ensure that the final regret is in terms of the unknown $\|u\|$.

Algorithm 5 should be seen as the analogue of the black-box reduction of Algorithm 1, but for bandit feedback instead of full information. The expected regret guarantee of Algorithm 5 is shown below, and the proof can be found in appendix C.

**Lemma 3.** *Suppose $\mathcal{A}_{\mathcal{V}}$ implements Interface 3 with $l \subseteq \mathbb{R}_+$, $w_t \in \mathcal{W}$ for all $t$, and let $L_{\mathcal{V}} = \max_t \partial \bar{\ell}_t(v_t)$. Suppose that $\ell_t(\mathbf{0}) = 0$. Then Algorithm 5 with $\delta, \alpha \in (0, 1]$ and $\eta = \sqrt{\frac{\delta^2}{4(dL)^2 T}}$ satisfies for all $\|u\| \in l$ and $r > 0$ with $\frac{ur}{\|u\|} \in \mathcal{Z}$,*

$$\mathbb{E}\left[\mathcal{R}_T(u)\right] = \widetilde{O}\left(1 + T\delta L \frac{\|u\|}{r} + \frac{\|u\|}{r}L_{\mathcal{V}}\sqrt{T} + \frac{\|u\|dL}{r\delta}\sqrt{T} + \alpha\|u\|_2 TL\right).$$

*In addition, if $\ell_t$ is also $\beta$-smooth for all $t$, then we have*

$$\mathbb{E}\left[\mathcal{R}_T(u)\right] = \widetilde{O}\left(1 + T\beta\delta^2\left(\frac{\|u\|}{r}\right)^2 + \frac{\|u\|}{r}L_{\mathcal{V}}\sqrt{T} + \frac{\|u\|}{r}\frac{dL}{\delta}\sqrt{T} + \alpha\|u\|_2 TL\right).$$

This bound has two main points not obviously under our direct control: the assumption that the $w_t$ lie in $\mathcal{W}$, and the value of $L_{\mathcal{V}}$, which is a bound on $|\partial \bar{\ell}_t(v_t)|$. In the remainder of this section we will specify the various settings of Algorithm 5 that guarantee that $w_t \in \mathcal{W}$ and that $L_{\mathcal{V}}$ is suitably bounded: two settings for the unconstrained setting and one for the constrained setting. The $\alpha\|u\|TL$ term is due to $z_t \in (1 - \alpha)\mathcal{Z}$ rather than $z_t \in \mathcal{Z}$, which induces a small amount of bias. The $r$ in Lemma 3 is to ensure that we satisfy the requirements for Online Gradient Descent to have a suitable regret bound. For unconstrained convex bandits $r = 1$. For constrained convex bandits we will see that $\frac{1}{r} = c$ (recall that we assume that $\frac{1}{c}\mathbb{B} \subseteq \mathcal{W} \subseteq \mathbb{B}$).

## 4.1 Unconstrained Convex Bandits

In this section we instantiate Algorithm 5 and derive regret bounds for either general convex losses or convex and smooth losses. We start with general convex losses. Since $\mathcal{W} = \mathbb{R}^d$, we do not need to

ensure that $\boldsymbol{z}_t + \delta\boldsymbol{s}_t \in \mathcal{W}$ and we can safely set $\alpha = 0$. This choice guarantees that $\boldsymbol{z}_t + \delta\boldsymbol{s}_t \in 2\mathbb{B}$ and that $|\partial\bar{\ell}_t(v_t)| \leq \frac{2dL}{\delta} + 2\delta L$. Then, Lemma 3 directly leads to Theorem 3 (the proof is deferred to appendix C.1).

**Theorem 3.** *Supppose $\mathcal{A}_\mathcal{V}$ implements Interface 3 with $l = \mathbb{R}_+$ and that $\ell_t(\mathbf{0}) = 0$. Then Algorithm 5 with $\delta = \min\{1, \sqrt{d}T^{-\frac{1}{4}}\}$, $\mathcal{Z} = \mathbb{B}$, $\alpha = 0$, and $\eta = \sqrt{\frac{\delta^2}{4(dL)^2 T}}$ satisfies for all $\boldsymbol{u} \in \mathbb{R}^d$,*

$$\mathbb{E}\left[\mathcal{R}_T(\boldsymbol{u})\right] = \widetilde{O}\left(1 + \|\boldsymbol{u}\|Ld\sqrt{T} + \|\boldsymbol{u}\|L\sqrt{d}T^{\frac{3}{4}}\right).$$

For unconstrained smooth bandits, we face an extra challenge. To bound the regret of Algorithm 5, $|\partial\bar{\ell}_t(v_t)| = |\langle \boldsymbol{z}_t, \hat{\boldsymbol{g}}_t\rangle + \beta 2\delta^2 v_t|$ must be bounded. Now in contrast to the linear or Lipschitz cases, in the smooth case $\ell_t(v_t)$ is not Lipschitz over $\mathbb{R}_+$. We will address this by artificially constraining $v_t$. Specifically, we ensure that $v_t \leq \frac{1}{\delta^3}$, which implies $|\delta^2 v_t| = O\left(\frac{1}{\delta}\right)$. This makes the Lipschitz constant of $\bar{\ell}_t$ to be dominated by the gradient estimate $\hat{\boldsymbol{g}}_t$ rather than the regularization. To see how this affects the regret bound, consider two cases, $\|\boldsymbol{u}\|_2 \leq \frac{1}{\delta^3}$ and $\|\boldsymbol{u}\|_2 > \frac{1}{\delta^3}$. If $\|\boldsymbol{u}\|_2 \leq \frac{1}{\delta^3}$ then we have not hurt anything by constraining $v_t$ since $\|\boldsymbol{u}\|_2$ satisfies the same constraint. If instead $\|\boldsymbol{u}\|_2 > \frac{1}{\delta^3}$ then the consequences for the regret bound are not immediately clear. However, following a similar technique in [8], we use the fact that the regret against $\mathbf{0}$ is $O(1)$ and the Lipschitz assumption to show that we have added a penalty of only $O(\|\boldsymbol{u}\|_2 LT)$:

$$\mathbb{E}[\mathcal{R}_T(\boldsymbol{u})] = \mathbb{E}[\mathcal{R}_T(\mathbf{0})] + \sum_{t=1}^T \mathbb{E}[\ell_t(\mathbf{0}) - \ell_t(\boldsymbol{u})] = O(1 + \|\boldsymbol{u}\|_2 LT).$$

Since $\|\boldsymbol{u}\|_2 > \frac{1}{\delta^3}$ the penalty for constraining $v_t$ is $O(\|\boldsymbol{u}\|_2 LT) = O(\|\boldsymbol{u}\|_2^2 L\delta^3 T)$, which is $O(\|\boldsymbol{u}\|_2^2 L\sqrt{T})$ if we set $\delta = O(T^{-1/6})$. The formal result can be found below and its proof can be found in appendix C.1.

**Theorem 4.** *Suppose $\mathcal{A}_\mathcal{V}$ implements Interface 3 with $l = (0, \frac{1}{\delta^3}]$, that $\ell_t$ is $\beta$-smooth for all $t$, and that $\ell_t(\mathbf{0}) = 0$. Then Algorithm 5 with $\delta = \min\{1, (dL)^{1/3}T^{-1/6}\}$, $\mathcal{Z} = \mathbb{B}$, $\alpha = 0$, and $\eta = \sqrt{\frac{\delta^2}{4(dL)^2 T}}$ satisfies for all $\boldsymbol{u} \in \mathbb{R}^d$,*

$$\mathbb{E}\left[\mathcal{R}_T(\boldsymbol{u})\right] = \widetilde{O}\left(1 + \max\{\|\boldsymbol{u}\|^2, \|\boldsymbol{u}\|\}\beta(dLT)^{\frac{2}{3}} + \max\{\|\boldsymbol{u}\|_2^2, \|\boldsymbol{u}\|\}dL^2\beta\sqrt{T}\right).$$

## 4.2 Constrained convex bandits

For the constrained setting we will set $\mathcal{Z} = \mathcal{W}$ and $\alpha = \delta$. This ensures that $v_t(\boldsymbol{z}_t + \delta\boldsymbol{s}_t) \in \mathcal{W}$ and we can apply Lemma 3 to find the regret bound in Theorem 5 below. Compared to the unconstrained setting, the regret bound now scales with $c$, which is due to the reshaping trick discussed in [13].

**Theorem 5.** *Suppose $\mathcal{A}_\mathcal{V}$ implements Interface 3 with $l = (0, 1]$ and that $\ell_t(\mathbf{0}) = 0$. Then Algorithm 5 with $\delta = \min\{1, \sqrt{d}T^{-1/4}\}$, $\mathcal{Z} = \mathcal{W}$, $\alpha = \delta = \min\{1, \sqrt{d}T^{-1/4}\}$, and $\eta = \sqrt{\frac{\delta^2}{4(dL)^2 T}}$ satisfies for all $\boldsymbol{u} \in \mathcal{W}$,*

$$\mathbb{E}\left[\mathcal{R}_T(\boldsymbol{u})\right] = \widetilde{O}\left(1 + (\|\boldsymbol{u}\|_2 + c\|\boldsymbol{u}\|)\sqrt{d}T^{3/4} + c\|\boldsymbol{u}\|dL\sqrt{T}\right).$$

# 5 Conclusion

In this paper, we develop the first algorithms that have comparator-adaptive regret bounds for various bandit convex optimization problems. The regret bounds of our algorithms scale with $\|\boldsymbol{u}\|$, which may yield smaller regret in favourable settings.

For future research, there are a number of interesting open questions. First, our current results do not encompass improved rates for smooth losses on constrained domains. At first blush, one might feel this is relatively straightforward via methods based on self-concordance [24], but it turns out that while such techniques provide good direction-learning algorithms, they may cause the gradients provided to the *scaling* algorithm to blow-up. Secondly, there is an important class of loss functions for which we did not obtain norm adaptive regret bounds: smooth and strongly convex losses. It is known that in this case an expected regret bound of $O(d\sqrt{T})$ can be efficiently achieved [17]. However, to achieve this regret bound the algorithm of Hazan and Levy [17] uses a clever exploration scheme, which unfortunately leads to sub-optimal regret bounds for our algorithms.

## Broader Impact

Our contribution is primarily theoretical, and we do not foresee any negative ethical or societal impact.

## Acknowledgments and Disclosure of Funding

Dirk van der Hoeven was supported by the Netherlands Organization for Scientific Research (NWO grant TOP2EW.15.211). Haipeng Luo was supported by NSF Awards IIS-1755781 and IIS-1943607. Ashok Cutkosky was employed at Google Research while working on this project.

## Footnotes

[1]Throughout the paper, the notation $\widetilde{O}$ hides logarithmic dependence on parameters $T$, $\|\boldsymbol{u}\|$, and $L$.

[2]For the linear case, this clearly holds since $\ell_t(\mathbf{0}) = 0$.

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
