[Supplementary Material]

## A  Details from section 2

*Proof of Lemma 1.* By definition we have

$$\mathcal{R}_T(\boldsymbol{u}) = \sum_{t=1}^{T} \langle \boldsymbol{w}_t - \boldsymbol{u}, \boldsymbol{g}_t \rangle = \sum_{t=1}^{T} \langle \boldsymbol{z}_t, \boldsymbol{g}_t \rangle (v_t - \|\boldsymbol{u}\|) + \|\boldsymbol{u}\| \sum_{t=1}^{T} \langle \boldsymbol{z}_t - \frac{\boldsymbol{u}}{\|\boldsymbol{u}\|}, \boldsymbol{g}_t \rangle$$

$$= \mathcal{R}_T^{\mathcal{V}}(\|\boldsymbol{u}\|) + \|\boldsymbol{u}\| \mathcal{R}_T^{\mathcal{Z}} \left( \frac{\boldsymbol{u}}{\|\boldsymbol{u}\|} \right).$$

$\square$

## B  Details from section 3

*Proof of Theorem 2.* For any fixed $\boldsymbol{u} \in \mathcal{W}$, let $r = \max_{\frac{r'\boldsymbol{u}}{\|\boldsymbol{u}\|} \in \mathcal{W}} r'$. Note that by definition we have $\frac{\|\boldsymbol{u}\|}{r} \in [0, 1]$ and $\frac{r\boldsymbol{u}}{\|\boldsymbol{u}\|} \in \mathcal{W}$. Therefore, similar to the proof of Lemma 1, we decompose the regret against $\boldsymbol{u}$ as:

$$\mathcal{R}_T(\boldsymbol{u}) = \sum_{t=1}^{T} \langle \boldsymbol{w}_t - \boldsymbol{u}, \boldsymbol{g}_t \rangle = \sum_{t=1}^{T} \langle \boldsymbol{z}_t, \boldsymbol{g}_t \rangle \left( v_t - \frac{\|\boldsymbol{u}\|}{r} \right) + \frac{\|\boldsymbol{u}\|}{r} \sum_{t=1}^{T} \langle \boldsymbol{z}_t - \frac{r\boldsymbol{u}}{\|\boldsymbol{u}\|}, \boldsymbol{g}_t \rangle,$$

which, by the guarantees of $\mathcal{A}_{\mathcal{V}}$ and $\mathcal{A}_{\mathcal{Z}}$,[3] is bounded in expectation by

$$\widetilde{O} \left( \frac{\|\boldsymbol{u}\|}{r} L\sqrt{T} + \frac{\|\boldsymbol{u}\|}{r} dL\sqrt{T} \right).$$

Finally noticing $\frac{1}{c} \leq r$ by the definition of $c$ finishes the proof. $\square$

## C  Details from section 4

*Proof of Lemma 3.* Denote by $\tilde{\boldsymbol{w}}_t = v_t \boldsymbol{z}_t$. By Jensen's inequality we have

$$\sum_{t=1}^{T} \mathbb{E}\left[\ell_t(\boldsymbol{w}_t) - \ell_t(\boldsymbol{u})\right] = \mathbb{E}\left[\sum_{t=1}^{T} \ell_t^{v_t}(\boldsymbol{w}_t) - \ell_t(\boldsymbol{u})\right] + \sum_{t=1}^{T} \mathbb{E}\left[\ell_t(\boldsymbol{w}_t) - \ell_t^{v_t}(\boldsymbol{w}_t)\right]$$

$$\leq \sum_{t=1}^{T} \mathbb{E}\left[\ell_t^{v_t}(\boldsymbol{w}_t) - \ell_t(\boldsymbol{u})\right]. \tag{5}$$

We now continue under the assumption that $\ell_t$ is $L$-Lipschitz. After completing the proof of the first equation of Lemma 3 we use the $\beta$-smoothness assumption to prove the second equation of Lemma 3.

Using the $L$-Lipschitz assumption we proceed:

$$\sum_{t=1}^{T} \mathbb{E}\left[\ell_t^{v_t}(\boldsymbol{w}_t) - \ell_t(\boldsymbol{u})\right] \leq \sum_{t=1}^{T} \mathbb{E}\left[\ell_t^{v_t}(\boldsymbol{w}_t) - \ell_t^{v_t}(\boldsymbol{u})\right] + \sum_{t=1}^{T} \mathbb{E}\left[\ell_t^{v_t}(\boldsymbol{u}) - \ell_t(\boldsymbol{u})\right]$$

$$\leq \sum_{t=1}^{T} \mathbb{E}\left[\ell_t^{v_t}(\boldsymbol{w}_t) - \ell_t^{v_t}(\boldsymbol{u})\right] + \mathbb{E}[L|v_t| \|\delta \boldsymbol{s}_t\|_2]$$

$$\leq \sum_{t=1}^{T} \mathbb{E}\left[\ell_t^{v_t}(\boldsymbol{w}_t) - \ell_t^{v_t}(\boldsymbol{u})\right] + \mathbb{E}[\delta L|v_t|]$$

$$= \sum_{t=1}^{T} \mathbb{E}\left[\ell_t^{v_t}(\tilde{\boldsymbol{w}}_t) - \ell_t^{v_t}(\boldsymbol{u})\right] + \mathbb{E}[\delta L|v_t|]$$

$$+ \sum_{t=1}^{T} \mathbb{E}\left[\ell_t^{v_t}(\boldsymbol{w}_t) - \ell_t^{v_t}(\tilde{\boldsymbol{w}}_t)\right]$$

$$\leq \sum_{t=1}^{T} \mathbb{E}\left[\ell_t^{v_t}(\tilde{\boldsymbol{w}}_t) - \ell_t^{v_t}(\boldsymbol{u})\right] + 2\,\mathbb{E}[\delta L|v_t|].$$

Now, by using the $L$-Lipschitz assumption once more we find that

$$\sum_{t=1}^{T} \mathbb{E}[\ell_t^{v_t}((1-\alpha)\boldsymbol{u}) - \ell_t^{v_t}(\boldsymbol{u})] \leq \alpha \|\boldsymbol{u}\|_2 TL \tag{6}$$

By using equation (6), the convexity of $\ell_t^{v_t}$, and Lemma 2 we continue with:

$$\sum_{t=1}^{T} \mathbb{E}\left[\ell_t(\boldsymbol{w}_t) - \ell_t(\boldsymbol{u})\right] \leq \sum_{t=1}^{T} \mathbb{E}\left[\langle \tilde{\boldsymbol{w}}_t - (1-\alpha)\boldsymbol{u}, \hat{\boldsymbol{g}}_t \rangle\right] + 2\,\mathbb{E}[\delta L|v_t|] + \alpha \|\boldsymbol{u}\|_2 TL$$

$$= \sum_{t=1}^{T} \mathbb{E}\left[\left(v_t - \frac{\|\boldsymbol{u}\|}{r}\right)\langle \boldsymbol{z}_t, \hat{\boldsymbol{g}}_t \rangle\right] + \mathbb{E}\left[\frac{\|\boldsymbol{u}\|}{r}\langle \boldsymbol{z}_t - \tilde{\boldsymbol{u}}, \hat{\boldsymbol{g}}_t \rangle\right]$$

$$+ \sum_{t=1}^{T} 2\,\mathbb{E}[\delta L|v_t|] + \alpha \|\boldsymbol{u}\|_2 TL$$

$$= \sum_{t=1}^{T} \mathbb{E}\left[\bar{\ell}_t(v_t) - \bar{\ell}_t\left(\frac{\|\boldsymbol{u}\|}{r}\right)\right] + \sum_{t=1}^{T} \frac{\|\boldsymbol{u}\|}{r}\,\mathbb{E}\left[\langle \boldsymbol{z}_t - \tilde{\boldsymbol{u}}, \hat{\boldsymbol{g}}_t \rangle\right]$$

$$+ 2T\delta L\frac{\|\boldsymbol{u}\|}{r} + \alpha \|\boldsymbol{u}\|_2 TL$$

where $\bar{\ell}_t(v) = v\langle \boldsymbol{z}_t, \hat{\boldsymbol{g}}_t \rangle + 2\delta L|v|$ as defined in Algorithm 5, $\tilde{\boldsymbol{u}} = \frac{r}{\|\boldsymbol{u}\|}(1-\alpha)\boldsymbol{u}$, and $r > 0$ is such that $\frac{\boldsymbol{u} r}{\|\boldsymbol{u}\|} \in \mathcal{Z}$.

Finally, by using the convexity of $\bar{\ell}_t$, plugging in the guarantee of $\mathcal{A}_{\mathcal{V}}$, and using Theorem 6 we conclude the proof of the first equation of Lemma 3:

$$\sum_{t=1}^{T} \mathbb{E}\left[\ell_t(\boldsymbol{w}_t) - \ell_t(\boldsymbol{u})\right]$$

$$\leq 2T\delta L\frac{\|\boldsymbol{u}\|}{r} + \mathbb{E}\left[\sum_{t=1}^{T}\left(v_t - \frac{\|\boldsymbol{u}\|}{r}\right)\partial \bar{\ell}_t(v_t)\right] + \frac{\|\boldsymbol{u}\|}{r}\,\mathbb{E}\left[\sum_{t=1}^{T}\langle \boldsymbol{z}_t - \tilde{\boldsymbol{u}}, \hat{\boldsymbol{g}}_t \rangle\right] + \alpha \|\boldsymbol{u}\|_2 TL$$

$$= \widetilde{O}\left(1 + T\delta L\frac{\|\boldsymbol{u}\|}{r} + \frac{\|\boldsymbol{u}\|}{r}L_{\mathcal{V}}\sqrt{T} + \frac{\|\boldsymbol{u}\|dL}{r\delta}\sqrt{T} + \alpha \|\boldsymbol{u}\|_2 TL\right).$$

Next, we continue from equation (5) under the smoothness condition. Using the definition of smoothness we find

$$\sum_{t=1}^{T} \mathbb{E}\left[\ell_t^{v_t}(\boldsymbol{w}_t) - \ell_t(\boldsymbol{u})\right] \leq \sum_{t=1}^{T} \mathbb{E}\left[\ell_t^{v_t}(\boldsymbol{w}_t) - \ell_t^{v_t}(\boldsymbol{u})\right] + \sum_{t=1}^{T} \mathbb{E}\left[\ell_t^{v_t}(\boldsymbol{u}) - \ell_t(\boldsymbol{u})\right]$$

$$\leq \sum_{t=1}^{T} \mathbb{E}\left[\ell_t^{v_t}(\boldsymbol{w}_t) - \ell_t^{v_t}(\boldsymbol{u})\right] + \mathbb{E}\left[\tfrac{1}{2}\beta |v_t|^2 \|\delta \boldsymbol{s}_t\|_2^2\right]$$

$$= \sum_{t=1}^{T} \mathbb{E}\left[\ell_t^{v_t}(\boldsymbol{w}_t) - \ell_t^{v_t}(\boldsymbol{u})\right] + \mathbb{E}\left[\tfrac{1}{2}\delta^2 |v_t|^2 \beta\right]$$

$$= \sum_{t=1}^{T} \mathbb{E}\left[\ell_t^{v_t}(\tilde{\boldsymbol{w}}_t) - \ell_t^{v_t}(\boldsymbol{u})\right] + \mathbb{E}\left[\tfrac{1}{2}\delta^2 |v_t|^2 \beta\right]$$

$$+ \sum_{t=1}^{T} \mathbb{E}\left[\ell_t^{v_t}(\boldsymbol{w}_t) - \ell_t^{v_t}(\tilde{\boldsymbol{w}}_t)\right]$$

$$\leq \sum_{t=1}^{T} \mathbb{E}\left[\ell_t^{v_t}(\tilde{\boldsymbol{w}}_t) - \ell_t^{v_t}(\boldsymbol{u})\right] + \mathbb{E}\left[\beta \delta^2 |v_t|^2\right].$$

Using equation (6), the convexity of $\ell_t^{v_t}$, and Lemma 2 we continue with:

$$\sum_{t=1}^{T} \mathbb{E}\left[\ell_t(\boldsymbol{w}_t) - \ell_t(\boldsymbol{u})\right]$$

$$\leq \sum_{t=1}^{T} \mathbb{E}\left[\langle \tilde{\boldsymbol{w}}_t - (1-\alpha)\boldsymbol{u}, \hat{\boldsymbol{g}}_t\rangle\right] + \mathbb{E}\left[\beta \delta^2 |v_t|^2\right] + \alpha \|\boldsymbol{u}\|_2 TL$$

$$= \sum_{t=1}^{T} \mathbb{E}\left[\left(v_t - \frac{\|\boldsymbol{u}\|}{r}\right)\langle \boldsymbol{z}_t, \hat{\boldsymbol{g}}_t\rangle\right] + \mathbb{E}\left[\beta \delta^2 |v_t|^2\right] + \sum_{t=1}^{T} \frac{\|\boldsymbol{u}\|}{r}\mathbb{E}\left[\langle \boldsymbol{z}_t - \tilde{\boldsymbol{u}}, \hat{\boldsymbol{g}}_t\rangle\right] + \alpha\|\boldsymbol{u}\|_2 TL$$

$$= T\beta\delta^2\left(\frac{\|\boldsymbol{u}\|}{r}\right)^2 + \sum_{t=1}^{T} \mathbb{E}\left[\bar{\ell}_t(v_t) - \bar{\ell}_t\left(\frac{\|\boldsymbol{u}\|}{r}\right)\right] + \sum_{t=1}^{T} \frac{\|\boldsymbol{u}\|}{r}\mathbb{E}\left[\langle \boldsymbol{z}_t - \tilde{\boldsymbol{u}}, \hat{\boldsymbol{g}}_t\rangle\right] + \alpha\|\boldsymbol{u}\|_2 TL,$$

where $\bar{\ell}_t(v) = v\langle \boldsymbol{z}_t, \hat{\boldsymbol{g}}_t\rangle + \beta\delta^2 v^2$ as defined in Algorithm 5. Finally, by using the convexity of $\bar{\ell}_t$, plugging in the guarantee of $\mathcal{A}_\mathcal{V}$, and using Theorem 6 we conclude the proof:

$$\sum_{t=1}^{T} \mathbb{E}\left[\ell_t(\boldsymbol{w}_t) - \ell_t(\boldsymbol{u})\right]$$

$$\leq T\beta\delta^2\left(\frac{\|\boldsymbol{u}\|}{r}\right)^2 + \mathbb{E}\left[\sum_{t=1}^{T}\left(v_t - \frac{\|\boldsymbol{u}\|}{r}\right)\partial\bar{\ell}_t(v_t)\right] + \frac{\|\boldsymbol{u}\|}{r}\mathbb{E}\left[\sum_{t=1}^{T}\langle \boldsymbol{z}_t - \tilde{\boldsymbol{u}}, \hat{\boldsymbol{g}}_t\rangle\right] + \alpha\|\boldsymbol{u}\|_2 TL$$

$$= \tilde{O}\left(1 + T\beta\delta^2\left(\frac{\|\boldsymbol{u}\|}{r}\right)^2 + \frac{\|\boldsymbol{u}\|}{r}L_\mathcal{V}\sqrt{T} + \frac{\|\boldsymbol{u}\|}{r}\frac{dL}{\delta}\sqrt{T} + \alpha\|\boldsymbol{u}\|_2 TL\right).$$

$\square$

**Theorem 6.** *Suppose that $\ell_t(\boldsymbol{0}) = 0$, that $\ell_t$ is $L$-Lipschitz for all $t$, and that $\mathcal{Z} \subseteq \mathbb{B}$. For $\boldsymbol{u} \in (1-\alpha)\mathcal{Z}$, Online Gradient Descent on $(1-\alpha)\mathcal{Z}$ with learning rate $\eta = \sqrt{\frac{\delta^2}{(dL)^2 4T}}$ satisfies*

$$\mathbb{E}\left[\sum_{t=1}^{T}\langle \boldsymbol{z}_t - \boldsymbol{u}, \hat{\boldsymbol{g}}_t\rangle\right] \leq 2\frac{dL}{\delta}\sqrt{T}.$$

*Proof.* The proof essentially follows from the work of Zinkevich [27], Flaxman et al. [13] and using the assumptions that $\ell_t(\boldsymbol{0}) = 0$ and that $\ell_t$ is $L$-Lipschitz. We start by bounding the norm of the

gradient estimate:

$$
\begin{aligned}
\|\hat{\boldsymbol{g}}_t\|_2 &= \frac{d}{v_t\delta}|\ell_t(\boldsymbol{w}_t)|\|\boldsymbol{s}_t\|_2 \\
&= \frac{d}{v_t\delta}|\ell_t(v_t(\boldsymbol{z}_t+\delta\boldsymbol{s}_t)) - \ell_t(\boldsymbol{0})| \\
&\leq \frac{dL\|\boldsymbol{z}_t+\delta\boldsymbol{s}_t\|_2}{\delta} \leq \frac{dL(1-\alpha+\delta)}{\delta}
\end{aligned}
\tag{7}
$$

By using equation (7) and the regret bound of Online Gradient Descent [27] we find that

$$
\begin{aligned}
\sum_{t=1}^{T}\langle\boldsymbol{z}_t,\hat{\boldsymbol{g}}_t\rangle - \min_{\boldsymbol{z}\in(1-\alpha)\mathcal{Z}}\sum_{t=1}^{T}\langle\boldsymbol{z},\hat{\boldsymbol{g}}_t\rangle &\leq \frac{(1-\alpha)}{2\eta} + \frac{\eta}{2}\sum_{t=1}^{T}\|\hat{\boldsymbol{g}}_t\|_2^2 \\
&\leq \frac{(1-\alpha)}{2\eta} + \frac{\eta}{2}\left(\frac{dL(1-\alpha+\delta)}{\delta}\right)^2 T \\
&\leq \frac{1}{2\eta} + 2\eta\left(\frac{dL}{\delta}\right)^2 T
\end{aligned}
$$

Plugging in $\eta = \sqrt{\frac{\delta^2}{(dL)^2 4T}}$ completes the proof. $\qquad\square$

## C.1 Details of section 4.1

*Proof of Theorem 3.* First, since $\ell_t(\boldsymbol{0}) = 0$, $\ell_t$ is $L$-Lipschitz, and $\boldsymbol{z}_t \in (1-\alpha)\mathcal{Z} = (1-\alpha)\mathbb{B}$ we have that

$$
\langle\boldsymbol{z}_t,\hat{\boldsymbol{g}}_t\rangle \leq \|\boldsymbol{z}_t\|_2\|\hat{\boldsymbol{g}}_t\|_2 \leq (1-\alpha)\frac{dL(1-\alpha+\delta)}{\delta} \leq \frac{2dL}{\delta},
\tag{8}
$$

where the first inequality is the Cauchy-Schwarz inequality and the second is due to equation (7). Since $|\partial\bar{\ell}_t(v_t)| \leq |\langle\boldsymbol{z}_t,\hat{\boldsymbol{g}}_t\rangle| + 2\delta L = L_{\mathcal{V}}$ we can use Lemma 3 to find

$$
\mathbb{E}[\mathcal{R}_T(\boldsymbol{u})] = \widetilde{O}\left(\delta TL\|\boldsymbol{u}\| + \|\boldsymbol{u}\|\frac{dL}{\delta}\sqrt{T} + \alpha TL\|\boldsymbol{u}\|_2\right).
$$

Plugging in $\alpha = 0$ and $\delta = \min\{1, \sqrt{d}T^{-\frac{1}{4}}\}$ completes the proof. $\qquad\square$

*Proof of Theorem 4.* By equation (8) $|\langle\boldsymbol{z}_t,\hat{\boldsymbol{g}}_t\rangle| \leq \frac{2dL}{\delta}$. Since $v_t \leq \frac{1}{\delta^3}$ we have that

$$
|\partial\bar{\ell}_t(v_t)| \leq \frac{dL}{\delta} + 2|v_t|\beta\delta^2 \leq \frac{dL+2\beta}{\delta} \leq \frac{\beta(dL+2)}{\delta}
$$

If $\|\boldsymbol{u}\|_2 \leq \frac{1}{\delta^3}$ applying Lemma 3 with $\alpha = 0$ gives us

$$
\mathbb{E}\left[\sum_{t=1}^{T}\ell_t(\boldsymbol{w}_t) - \ell_t(\boldsymbol{u})\right] = \widetilde{O}\left(1 + T\beta\delta^2\|\boldsymbol{u}\|^2 + \|\boldsymbol{u}\|\frac{dL\beta}{\delta}\sqrt{T}\right).
\tag{9}
$$

If $\|\boldsymbol{u}\|_2 > \frac{1}{\delta^3}$ then using the Lipschitz assumption on $\ell_t$ and equation (9) with $\boldsymbol{u} = \boldsymbol{0}$ gives us

$$
\begin{aligned}
\mathbb{E}\left[\sum_{t=1}^{T}\ell_t(\boldsymbol{w}_t) - \ell_t(\boldsymbol{u})\right] &= \mathbb{E}\left[\sum_{t=1}^{T}\ell_t(\boldsymbol{w}_t) - \ell_t(\boldsymbol{0}) + \ell_t(\boldsymbol{0}) - \ell_t(\boldsymbol{u})\right] \\
&= \widetilde{O}(1 + \|\boldsymbol{u}\|_2 LT) \\
&= \widetilde{O}(1 + \|\boldsymbol{u}\|_2^2\delta^3 LT),
\end{aligned}
\tag{10}
$$

where we used that $\|\boldsymbol{u}\|_2 \geq \frac{1}{\delta^3}$. Adding equations (9) and (10) gives

$$
\mathbb{E}\left[\sum_{t=1}^{T}\ell_t(\boldsymbol{w}_t) - \ell_t(\boldsymbol{u})\right] = \widetilde{O}\left(1 + \|\boldsymbol{u}\|_2^2\delta^3 LT + T\beta\delta^2\|\boldsymbol{u}\|^2 + \|\boldsymbol{u}\|\frac{\beta dL}{\delta}\sqrt{T}\right)
$$

Setting $\delta = \min\{1, (dL)^{1/3}T^{-1/6}\}$ gives us

$$\mathbb{E}\left[\sum_{t=1}^{T} \ell_t(\boldsymbol{w}_t) - \ell_t(\boldsymbol{u})\right] = \widetilde{O}\left(1 + \max\{\|\boldsymbol{u}\|^2, \|\boldsymbol{u}\|\}\beta(dLT)^{\frac{2}{3}} + \max\{\|\boldsymbol{u}\|_2^2, \|\boldsymbol{u}\|\}dL^2\beta\sqrt{T}\right).$$

$\square$

## C.2 Details of section 4.2

*Proof of Theorem 5.* First, to see that $\boldsymbol{z}_t + \delta\boldsymbol{s}_t \in \mathcal{W}$ recall that by assumption $\mathcal{W} \subseteq \mathbb{B}$. Since $\alpha = \delta$ we have that $\boldsymbol{z}_t + \delta\boldsymbol{s}_t \in (1-\alpha)\mathcal{W} + \delta\mathbb{S} \subseteq (1-\delta)\mathcal{W} + \delta\mathcal{W} = \mathcal{W}$. For any fixed $\boldsymbol{u} \in \mathcal{W}$, let $r = \max_{\frac{r'\boldsymbol{u}}{\|\boldsymbol{u}\|}\in\mathcal{W}} r'$. Note that by definition we have $\frac{\|\boldsymbol{u}\|}{r} \in [0, 1]$ and $\frac{r\boldsymbol{u}}{\|\boldsymbol{u}\|} \in \mathcal{W}$. By using equation (8) we can see that $|\partial\bar{\ell}_t(v_t)| \leq \frac{dL}{\delta} + 2\delta L$. By definition, $\frac{1}{r} \leq c$. This implies that the regret of $\mathcal{A}_\mathcal{V}$ is $\widetilde{O}\left(1 + \frac{\|\boldsymbol{u}\|}{r}\frac{dL}{\delta}\sqrt{T}\right)$. Applying Lemma 3 with the parameters above we find

$$\mathbb{E}\left[\sum_{t=1}^{T} \ell_t(\boldsymbol{w}_t) - \ell_t(\boldsymbol{u})\right] = \widetilde{O}\left(1 + (\|\boldsymbol{u}\|_2 + c\|\boldsymbol{u}\|)TL\delta + c\|\boldsymbol{u}\|\delta L\sqrt{T} + c\|\boldsymbol{u}\|\frac{dL}{\delta}\sqrt{T}\right).$$

Finally, setting $\delta = \min\{1, \sqrt{d}T^{-1/4}\}$ completes the proof:

$$\mathbb{E}\left[\sum_{t=1}^{T} \ell_t(\boldsymbol{w}_t) - \ell_t(\boldsymbol{u})\right] = \widetilde{O}\left(1 + (\|\boldsymbol{u}\|_2 + c\|\boldsymbol{u}\|)\sqrt{d}T^{3/4} + c\|\boldsymbol{u}\|dL\sqrt{T}\right).$$

$\square$

## Footnotes

[3]Note that the condition $|\langle z_t, g_t \rangle| \leq 1$ in Algorithm 4 indeed holds in this case since $\mathcal{Z} = \mathcal{W} \subseteq \mathbb{B}$ and $\|g_t\|_2 \leq L$ by the Lipschitzness condition.