[Reviews · NeurIPS 2020]

Review 1

Summary and Contributions: This paper provides the first regret bound for convex bandit problems with the comparator-adaptive guarantee. Concretely, the regret bound will scale with the magnitude of the comparator, which is the optimal point. In some circumstances, such the comparator-adaptive guarantee might be much better than the traditional regret bound which is independent of the comparator.

Strengths: The paper is clearly written and the proof is simple and clean. The key idea of reducing the bandit setting to the full-information setting is cute and might be of interest to other applications.

Weaknesses: It could be more convincing on why people should care about the comparator-adaptive guarantee if the author could provide some examples showing the separation of being comparator-adaptive or not.

Correctness: The main idea of the paper is to reduce the bandit setting to the full-information setting where previous work provided the comparator-adaptive guarantee. The reduction is cleanly presented in Section 2 and 3. As I don't get enough time to go through the whole supplementary materials, I think the technical content should be correct modulo minor errors.

Clarity: The paper is nicely written with a good introduction and clean technical content. Especially, I like the analogy/example in the first paragraph of the introduction and I'm sure that I'll never choose not to eat anything at all.

Relation to Prior Work: The author provides references for previous work on the comparator-adaptive guarantee for the full-information setting and related works in the bandit setting.

Reproducibility: Yes

Additional Feedback: After the response phase, considering the additional feedback received, I remain with my initial assessment of the paper.


Review 2

Summary and Contributions: This paper considers the bandit convex optimization problem, and the goal is to design algorithms that can adapt to the norm of the unknown comparator $u$, instead of its maximal norm. By doing so, the algorithm can enjoy more adaptivity and hold in the unconstrained learning scenario. This paper follows the black-box reduction method of Cutkosky and Orabona [10] to separately learn the magnitude and the direction. For the bandit convex optimization problem, some extra efforts are requeired. The major novelty of this paper in my opinion is to introduce a novel surrogate loss for the scaling algorithm, speficially designed for the BCO problem.

Strengths: The paper has a clear motivation of designing algorithms to adapt the comparator norm, which is in the line of parameter-free online learning. + Previous results, as far as I know, only consider full-information setting. This paper studies the BCO setting. + The paper mainly follows the idea of black-box reduction method of Cutkosky and Orabona [10] to separately learn the magnitude and the direction. But to do BCO, some extra efforts are required. In particular, a new surrogate loss in terms of $v$ is required for the scaling algorithm $\mathcal{V}$. Its design is carefully specified for the problem. + The paper writing is clear.

Weaknesses: - The work heavily builds on the previous work of Cutkosky and Orabona [10]. In particular, the power of the proposed BCO method really relies on the paramter-free nature of the scale learning interface in "Interface 3". So this diminishes its significance from the side of technical contributions. - In line 72, the authors argue two new ideas "appropriate surrogate loss function" and "a new one-point gradient estimator with time-varying parameters". I admit that the design of surrogate loss is clever, while the gradient estimator seems standard, with only a slight and nature twist for the problem. So it is not appropriate to emphasize much on that.

Correctness: As far as I have checked, the technical proofs are correct.

Clarity: Yes, this paepr is well written and well structured.

Relation to Prior Work: The paper mainly builds on the work of Cutkosky and Orabona [10], and the authors have acknowledged that properly and clearly. Related work section (line 76-81). There are recently several papers [1,2] consider the BCO setting with non-stationary regret analysis, which can be regarded as an adaptivity to the non-stationary environment. In particular, [2] also introduce the surrogate loss for BCO problem to achieve non-stationary regret bounds. [1] https://arxiv.org/abs/1907.13616 [2] https://arxiv.org/abs/1907.12340

Reproducibility: Yes

Additional Feedback:


Review 3

Summary and Contributions: In this paper, the authors study bandit convex optimization methods that adapt to the norm of the comparator. They use and extend techniques from the full-information setting to develop comparator-adaptive algorithms for linear bandits and convex bandits. The regret bounds of their algorithms are small whenever the norm of the comparator is small.

Strengths: 1. The setting and results in this paper new. This is the first paper that establish comparator-adaptive regret bounds for the bandit setting. 2. The paper considers several types of functions, including linear, convex, and convex and smooth.

Weaknesses: 1. The algorithms are natural extensions of previous comparator-adaptive algorithms for the full-information setting. Although the authors have introduced necessary modifications, such as the smoothing in (2) and the surrogate losses in Algorithm 5, the novelty is somehow limited. There is indeed one challenge for the bandit setting, e.g. the $1/v$ issue in Line 199, but is was addressed by an extra assumption instead of a more advanced algorithm. 2. The upper bounds exhibit various dependences on $c$ and $d$. It is not clear whether those dependences are tight.

Correctness: Yes

Clarity: Yes

Relation to Prior Work: Yes

Reproducibility: Yes

Additional Feedback: From the discussions from Line 255, it seems that the assumption $\ell_t(0)=0$ is only used in Theorem 4. Is it true? Typo: 1. The self-concordant parameter $\nu$ did not appear in Table 1, although it was introduced in the title. --------after rebuttal---------- The rebuttal addressed some of my concerns regarding the novelty. So, I increase my score to 6.

[Author Response · NeurIPS 2020]

Dear reviewers and chairs,

We thank all reviewers for their careful reading of the manuscript.

Reviewer # 1:

We will try to add more examples on when it helps to be comparator-adaptive. Note that in the paper we already have a
nice but subtle application of the comparator-adaptive property in line 256, where we use that the regret against $\mathbf{0}$ is
$O(1)$. Without using this property the analysis of the unconstrained smooth case actually becomes troublesome as the
surrogate losses for the unconstrained smooth case are not Lipschitz on $[0, \infty)$.

Reviewer # 2:

First, thank you for the references. Especially the the second reference appears to be related as it seems to extend the
ideas of MetaGrad to the Bandit Convex Optimization setting.

As for our technical contribution, understanding how to utilize the ideas of Cutkosky and Orabona [10] in the Bandit
Convex Optimization setting is one of the main contributions of this paper. There are subtle but important differences
between the full-information setting and the Bandit Convex optimization setting that require us to make (not so) subtle
adaptations to be able to apply the ideas of Cutkosky and Orabona [10]. For example, as we point out in section
3.2, the projections of Cutkosky and Orabona [10] do not directly work in the bandit setting. Another example is the
gradient estimator. Even though it appears as a natural estimator some care is required to be able to use it as it is tightly
connected to the surrogate losses we feed to Interface 3. However, in the final version we will reduce the emphasis on
the gradient estimator as a stand-alone novelty. Also note that there is an entire line of work that expands the ideas of
Cutkosky and Orabona [10] to other settings, see for example [8], [20], and [26].

Reviewer # 3:

Even though the extensions indeed appear to be natural, as reviewer # 1 does we would like to argue that this is a
strength rather than a weakness. One of the main contributions is that our techniques represent a way to utilize the
ideas of Cutkosky and Orabona [10] in the Bandit Convex Optimization setting and we hope that our insights will allow
future researchers to design further extensions. As reviewer # 1 points out, the way we extend the ideas of Cutkosky
and Orabona [10] to the Bandit Convex Optimization setting may be of interest to other applications. With simple and
clean proofs the ideas will be easier to use by other researchers, otherwise our hope could be in vain as the ideas might
be too obfuscated to be of use.

Regardless, there are other subtle modifications that one may miss. For example, as we mentioned in our comments to
reviewer # 1, in the unconstrained smooth case the surrogate losses are not Lipschitz on $[0, \infty)$, which means that a
direct application of Interface 3 would not work. Instead, we restrict the algorithm to play on a smaller domain where
the surrogate losses are Lipschitz. This leaves us with a problem if the comparator is outside of this smaller domain, but,
inspired by [8], we manage to solve this by utilizing the fact that our comparator-adaptive algorithm has $O(1)$ regret
against $\mathbf{0}$.

We would also like to address our assumption of zero loss at zero, which is helpful for overcoming one of the key
challenges mentioned in Line 199. While it's obviously best to have as few assumptions as possible, this particular
assumption actually holds in many practical scenarios, as we discuss in the paper. Moreover, finding improved
performance on special problem classes can open up interesting new research directions - for example, it may even be
that an assumption similar to ours is actually *necessary* in order to obtain the comparator-adaptive bounds!

As for the dependencies on $c$, if the domain is a unit ball (in any norm) then the dependency on $c$ disappears as it is 1 in
that case. For more general domains it is not clear if the dependency on $c$ is tight. To our knowledge, the only place in
literature where a similar constant appears in the regret bounds is in Flaxman et. al. (2005) [13], which was removed in
subsequent work. Because of this we suspect that, as in the case where the domain is a unit ball, the optimal regret
bound should not depend on $c$. As for the dependency on $d$, it is tight in the linear case (see Dani et. al. (2008) [11]).

Finally, to clarify, the assumption $\ell_t(0) = 0$ is used in Theorems 3, 4, and 5. Thank you for pointing out the typo in
Table 1.

[Meta-Review · NeurIPS 2020]

The reviewers agree that the paper is clean and well-written, and the results are novel and correct. The reviewers found the main contribution as the extension of the techniques in Cutkosky and Orabona [10].for full-information setting to the bandit convex optimization techniques, which requires subtle, though not substantially new, extension to their techniques.